# HPLC MS-MS Analysis Shows Measurement of Corticosterone in Egg Albumen Is Not a Valid Indicator of Chicken Welfare

**DOI:** 10.3390/ani10050821

**Published:** 2020-05-09

**Authors:** Malcolm P. Caulfield, Matthew P. Padula

**Affiliations:** School of Life Sciences and Proteomics Core Facility, Faculty of Science, The University of Technology Sydney, Ultimo 2007, Australia; matthew.padula@uts.edu.au

**Keywords:** animal welfare, chickens, layer hens, stress hormones, corticosterone, corticosteroids, mass spectrometry, high-pressure liquid chromatography

## Abstract

**Simple Summary:**

Growing interest in the welfare of farmed animals, particularly those in restrictive housing, has stimulated attempts to define simple measures of welfare. One such claimed measure involves the analysis of so-called stress hormones. In chickens, the main ‘stress hormone’ is corticosterone. In Australia, reviews of legislation relating to the welfare of chickens housed in cages have relied heavily on non-invasive measures of corticosterone, in particular those using egg white (albumen). All of those measures have used antibodies to quantify the corticosterone. Recently, doubts have been raised about the specificity of these measurement techniques. In this study, we demonstrate that high-resolution chromatographic separation of extracted egg albumen, followed by mass spectrometry, reveals that corticosterone is barely detectable in chicken egg albumen. Previous work using immunoassays reported levels of 0.5 to 20 ng/g. We have found egg albumen corticosterone concentrations of about 50 pg/g. We conclude there is so little corticosterone in egg albumen that it is not routinely usable as an indicator of chicken welfare. We have also found significant amounts of other steroids (progesterone, cortisol) in chicken egg white, which may have contributed to the levels reported in the antibody studies.

**Abstract:**

Assessment of animal welfare can include analysis of physiological parameters, as well as behavior and health. Levels of adrenocortical hormones such as cortisol (and corticosterone in chickens) have been relied on as indicators of stress. Elevations in those hormones have been said to be correlated with poor welfare, while levels in the normal range have been interpreted to mean that animals are in a good state of welfare. Procuring blood samples from animals for hormone measures can in itself be stressful and cause increases in the target hormones. To overcome this problem, indirect measures of cortisol and corticosterone have been developed. In chickens, corticosterone levels in egg albumen are said to be a useful indirect measure, and have been used in several recent studies as indicators of chicken welfare. All of the measures of chicken egg albumen corticosterone in welfare studies have used immunoassays, and have reported values ranging from about 0.5 to over 20 ng/g. Using these measures, egg albumen from chickens housed in conventional cages or free ranging has been said to have indistinguishable corticosterone levels. This has been used to support the conclusion that chickens kept in conventional cages are not experiencing stress and are in a good state of welfare. In this study, we have used high-pressure liquid chromatography (HPLC) coupled with mass spectrometry (MS) to measure corticosterone in egg albumen. We found levels of corticosterone (median level about 50 pg/g) in egg albumen which were just above the limit of detection. By contrast, we found significant levels of progesterone and cortisol, hormones which would be expected to cross react with anti-corticosterone antibodies, and which therefore might explain the high reported levels of corticosterone using immunoassay. We conclude that because corticosterone levels in egg albumen are negligible, they cannot be used as an indicator of chicken welfare.

## 1. Introduction

There is increasing public concern about close confinement of animals in intensive farming systems. One such system is the conventional cage for egg-laying chickens. Nicol and colleagues recently provided a useful description of conventional cages and free range systems for layer hens, and a summary of studies of factors affecting animal welfare in those systems [1]. It is accepted that assessment of animal welfare should involve a range of techniques, including behavioural, clinical (i.e., health) and physiological. Thus, there are said to be three ‘conceptual frameworks’ which can be applied: affective state, natural living and biological functioning [2]. An extensively used physiological measure (under the ‘biological functioning’ framework) is the assay of steroid effector hormones released upon activation of the hypothalamic–pituitary–adrenal (HPA) axis [3]. It is said that stressors cause elevation of adrenal corticosteroids (principally corticosterone in chickens [4]), and that because stress results in poor welfare, raised levels of corticosterone indicate a poor welfare state. The uncritical use of corticosteroid levels as an indicator of animal welfare has long been the subject of criticism [5], particularly for different egg-laying chicken housing systems. Different systems have been reported as being associated with increases, decreases or no difference in corticosterone levels [1,5]. 

The frequent use of corticosteroid measurements in animal welfare studies in recent decades has been facilitated by the availability of antibodies for the target molecules, and the resultant wide range of commercially available immunoassay kits. Most of the early studies of corticosteroids used plasma. However, it is apparent that the restraint of an animal to take a blood sample, and the act of taking the sample, will itself cause increases in the measured steroid. This has prompted efforts to develop non-invasive techniques for steroid measurement, including corticosterone, in chickens [6,7]. In particular, measures of faecal metabolites of corticosterone, using hplc and immunoassay, may have promise as an indirect measure [7]. One widely used measure in egg-laying chickens is the direct (i.e., without chromatographic separation) immunoassay of corticosterone in egg albumen [8]. Equivalent levels of egg albumen corticosterone in chickens that are housed free-range or in battery cages [9] have been used to support the idea that free-range hens are not necessarily less stressed than cage hens [10]. Further studies have applied egg corticosterone as an indicator of chicken welfare in studies of free-range chickens [11], ‘furnished cages’ [12] and floor space allowance and nest box access [13,14]. 

The use of direct immunoassays to measure steroids in human medicine, and in areas such as sports doping, has been heavily criticized, primarily because anti-steroid antibodies lack specificity, with significant and confounding cross-reactivity with related molecules. It is now accepted that mass spectrometry methods represent the ‘gold standard’ for steroid measurements [15]. Regarding measures in eggs, problems with the cross-reactivity of anti-corticosterone antibodies were highlighted in a study of bird egg yolk, where high-performance liquid chromatography (HPLC) separation, followed by immunoassay, indicated that the dominant immunoreactivity occurred at a peak corresponding to progesterone and similar molecules. There was very little corticosterone [16]. Consistent with this, a mass spectrometric analysis of chicken egg albumen indicated that corticosterone was barely detectable (at just under 50 pg/g) [17].

The aim of the present study was to develop a method for measuring corticosterone and similar steroid molecules in egg albumen using HPLC coupled with mass spectrometry. 

## 2. Materials and Methods 

All solvents (methanol, acetonitrile, isopropanol, hexane and ethyl acetate) were of HPLC grade (B&J) and were obtained from ChemSupply. MS (mass spectrometry) grade formic acid (Fluka) was obtained from ChemSupply. Water was purified to MS standard by carbon adsorber and mixed-bed ion exchange resins (Sartorius). To avoid problems with steroids binding to glass [18], all procedures were carried out in plasticware. Autosampler vials were polypropylene microvials (Shimadzu). Solid phase extraction (SPE) was done using C-18 50 mg columns (Discovery DSC-18) from Merck. The following compounds were obtained from Merck: corticosterone, cortisol, 11-deoxycortisol, 11-deoxycorticosterone and progesterone. Deuterated corticosterone (d8-corticosterone) for use as internal standard (2,2,4,6,6,17A,21,21-D8, 97%–98%) was obtained from Cambridge Isotope Laboratories. 

Figure 1 shows some of the major synthetic and metabolic pathways for the steroids studied.

Standards were prepared as a 1 mg/mL solution in methanol. A mix of corticosterone, cortisol, 11-deoxy cortisol, 11-deoxy corticosterone, progesterone and d8-corticosterone was prepared in 50% methanol/water in concentrations ranging from 0.02 to 4 ng/mL.

Twelve conventional cage eggs from hens of the Isa Brown strain were kindly supplied by Dr Tamsyn Crowley, School of Environmental and Rural Science, University of New England, Armidale, New South Wales, Australia. They were shipped overnight to the laboratory in Sydney, where albumen was separated and stored at −80 °C until use. All eggs were collected within a few hours of lay. Mouse serum was kindly supplied by Dr Fiona Ryan, University of Technology, Sydney.

Albumen samples were thawed overnight at −4 °C, then homogenized on ice using a Heidolph Diax 600 homogenizer, with four bursts of 30 s each. The criterion for adequate homogenization was that the samples could be easily pipetted with a 1 mL pipette tip.

Samples of albumen (1 mL = 1 g) were pipetted into a plastic 15 mL Falcon tube, and 40 µL d8-corticosterone 0.125 µg/mL was added to give a final concentration of internal standard of 0.5 ng/g. The extraction method was based on that of De Baere et al. [15], incorporating liquid–liquid extraction with protein precipitation, defatting with hexane and further purification on SPE columns. Acetonitrile (5 mL) containing 1% formic acid was added to each spiked 1mL sample and vortexed for 30 s. The samples were centrifuged for 10 min at 5000 rpm at 4 °C. A total of 2.5 mL supernatant was taken, and 2.5 mL hexane was added, and the tubes were vortexed for 15 s. The upper hexane layer was discarded. Acetonitrile was evaporated with a stream of nitrogen at 40 °C and the residue resuspended in 100 µL methanol, which was vortexed for 30 s, and 900 µL water was added. SPE columns were conditioned with the addition of 1 mL methanol, followed by l mL water, after which the sample was added. After the sample had passed through the column, the column was washed with 1 mL water, followed by 1 mL hexane. Samples were eluted with two 1 mL volumes of ethyl acetate. Elution of all solvents was by gravity. The eluate was dried under nitrogen gas at 40 °C and resuspended in 125 µL methanol, vortexed for 30 s, followed by the addition of 125 µL water. The resulting sample was centrifuged at 10,000 rpm (4 °C) for 10 min and 100 µL of the supernatant was added to autosampler vials for HPLC-MS-MS analysis. The autosampler was maintained at 5 °C. Studies using mouse serum as a positive control for corticosterone extracted a serum volume of 100 µL. All other extraction and analysis steps were as for egg albumen. A total of 10 µL volumes of samples were injected onto the HPLC column.

The HPLC–MS system was based on a Shimadzu LCMS 8060 triple quadrupole mass spectrometer, with two Nexera LC30AD pumps, a CTO-20AC column oven and a SIL-30AC autosampler. The HPLC column was an Agilent Zorbax Eclipse Plus C18 column, having dimensions 2.1 × 100 mm packed with 1.8 µm particles, maintained in the column oven at 40 °C. Mobile phase A was 0.1% formic acid in water, and mobile phase B was acetonitrile plus 0.1% formic acid. Flow rate was 0.4 mL/min. Two gradients were used (Figure 2).

Gradient 1 was used in preliminary experiments and was adequate for the separation of all the studied steroids, apart from corticosterone. Better separation of corticosterone from other peaks with similar retention times was achieved with Gradient 2, and this was used for quantification of corticosterone in samples. Because of the length of the gradients (15 min), retention times shifted by about 0.2 min during the 8-h analysis time, which was necessary to process all the samples and standards. To allow for this, standards at 0.1 ng/mL were interposed at every 5th sample. Retention time data from those standards were used to adjust the retention times for corticosterone peaks in the samples.

Table 1 shows the MS parameters for precursor ions and fragment ions, together with observed retention times, for the steroids used in the present study. 

All other parameter settings for the mass spectrometer were as previously described [19]. Dwell time for transitions was 50 ms. Data were acquired and analysed with Shimadzu LabSolutions v5.91 software. The values reported for each steroid are integrated peak counts for the quantifier ion and are shown in Table 1. GraphPad Prism 8.3.1 was used for statistical analysis and graphing.

## 3. Results

Traces obtained for the corticosterone standard curve and a plot of those traces are shown in Figure 3. The limit of quantification was determined to be 0.02 ng/mL (Figure 3a inset).

Figure 4 shows overlaid MS traces for albumen extracts, with corticosterone peaks (filled arrows), and other peaks at around the retention time for corticosterone, obtained with Gradient 1 (a) and Gradient 2 (b), using albumen samples from 12 cage eggs. 

While Gradient 1 clearly separated peaks for corticosterone and the isobaric molecule 11-deoxy cortisol, it was apparent there was at least one earlier peak at almost the same retention time for corticosterone, which would interfere with the integration of peaks at the apparent corticosterone retention time. Samples run with Gradient 2 showed clearer corticosterone peaks (Figure 4b), and Gradient 2 was used for measurement of corticosterone peaks. It is important to note that, with the exception of one reading, even using Gradient 2, almost all corticosterone peaks were just above the level of detection.

Gradient 1 gave good separation of all the other studied steroids (Figure 5) and was used for the quantification of steroids other than corticosterone in samples (Figure 6).

Standard lines obtained for steroids other than corticosterone were linear in the range of concentrations studied, and were fitted with a straight line. R^2^ values for all regressions were greater than 0.998 (data not shown). 

Mean recovery, calculated from levels of d8 corticosterone in samples, was 48.1%. The 95% confidence interval for recoveries was from 44% to 52.2%. Figure 7 shows individual datapoints for corticosterone and the other studied steroids. In our opinion, the scatter of datapoints was inconsistent with a normal distribution. It seemed appropriate, therefore, to use the median of all data to represent central tendency, and to calculate *p* values for comparison of cage and free-range albumen samples using a non-parametric test (Mann–Whitney). 

Steroid concentrations in egg albumen samples are shown in Table 2. Corticosterone levels were very low, as was 11-deoxycorticosterone. There were significant quantities of 11-deoxycortisol, cortisol and progesterone. Mean (n = 6) corticosterone levels in mouse serum were 71 ng/mL, ranging from 49 to 80 ng/mL, which is similar to earlier reported values using HPLC and MS [20]. 

## 4. Discussion

It is a problem besetting animal welfare science that the measurement of ‘stress hormones’ is ‘dangerously easy’ [21]. This issue has been acknowledged and addressed in human clinical chemistry [15], but has yet to have any impact on the routine measurement of cortisol and corticosterone in animal welfare studies.

The present study, using HPLC and mass spectrometry, has demonstrated that corticosterone levels in chicken egg albumen are just above the threshold of detectability. Median values of corticosterone were less than 50 pg/g. These values are consistent with those found in commercially bought eggs in an earlier HPLC–mass spectrometry study [17]. In albumen samples, we found unidentified peaks with indistinguishable precursor and fragment masses, and similar retention times to corticosterone. These peaks were better separated from the corticosterone peak by using an extended chromatographic gradient. We also found that chicken egg albumen contains significant quantities of progesterone (about 1.2 ng/g) and cortisol (about 0.4 ng/g), with detectable, but lower, levels of 11-deoxycortisol and 11-deoxycorticosterone. 

These data show that earlier studies of egg albumen using immunoassay probably greatly overestimated corticosterone content, by up to 20-fold, given the cross-reactivity of anti-corticosterone antibodies for closely related steroids such as cortisol and progesterone [16,22]. This will have been compounded by the presence of other unidentified molecules, which may have similar structures to corticosterone and which may interact with the antibodies used. Our findings go some way to explaining the greater than 50-fold range of egg albumen corticosterone levels reported using immunoassays, even in separate studies from the same laboratory, which employed both radioimmunoassay and enzyme-linked immunosorbent assay [13,14].

The main conclusion from this work is that immunoassays of corticosterone in chicken egg albumen cannot be used as measures of chicken welfare. The bulk of what is measured is probably not corticosterone. This is perhaps not surprising, given that corticosterone must move from the plasma to albumen during the few hours in which it is formed [23], and is present in chicken plasma at relatively low levels (about 0.5 ng/mL: [24]).

Similarities between egg corticosterone levels in chickens housed in different housing systems, particularly conventional cages and free ranging, continue to be used to support the contention that chickens housed in cages have good welfare [10]. Given that it is likely that corticosterone is not what is being measured, such conclusions are arguably questionable. 

Our finding that there is almost undetectable corticosterone in chicken egg albumen adds to the substantial list of criticisms of the assay of ‘stress hormones’ in animal welfare, even where those measures do not involve blood sampling. Thus, adrenocortical hormones undergo significant natural diurnal variation, with pulsatile rises and falls superimposed on that rhythm [25]. Given this, measures at one timepoint are unlikely to reflect relevant levels. Moreover, given the substantial variations in levels observed, there is the obvious question as to the significance of those variations [5]. They are unlikely to represent diurnal variations in welfare. Exposure to sustained stress results in a down-regulation of the adrenocortical response, making measures of these hormones unsuitable for anything other than acute stress situations [3]. This has particular importance where the question is the effect of long-term housing on welfare. A further serious criticism of reliance on cortisol or corticosterone measures is that many situations which are demonstrably stressful (and associated with marked changes in other welfare measures) are not associated with increases in those hormones [5,26,27,28]. Furthermore, it is clear that corticosteroids can be elevated in response to arousal, as well as stress [1], requiring the ‘valence’ of the response to be defined before anything can be concluded about stress or animal welfare. 

The conclusion is that, unless corticosterone (or other ‘stress hormone’ measures) can be validated and verified as measures of welfare, they should be discounted in favour of other measures, including behavioural, ethological and health measures, all of which have been applied extensively in the area of chicken welfare [29]. This is particularly the case where the assessment of welfare is relied on as part of the legislative process relating to the implementation of animal welfare law [30]. 

## Figures and Tables

**Figure 1 animals-10-00821-f001:**
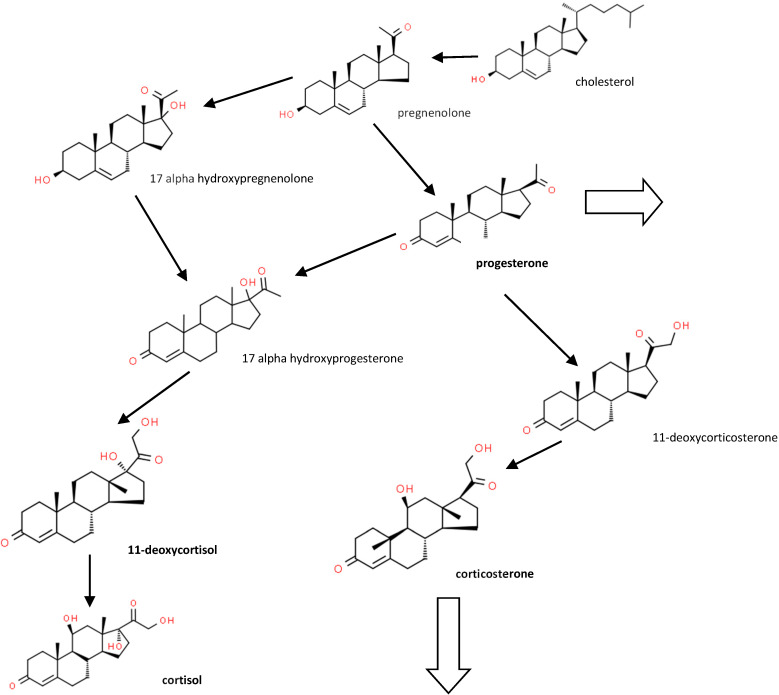
Major corticosteroid metabolic pathways (see [4] for a more detailed pathway). Large arrows show pathways to other metabolites.

**Figure 2 animals-10-00821-f002:**
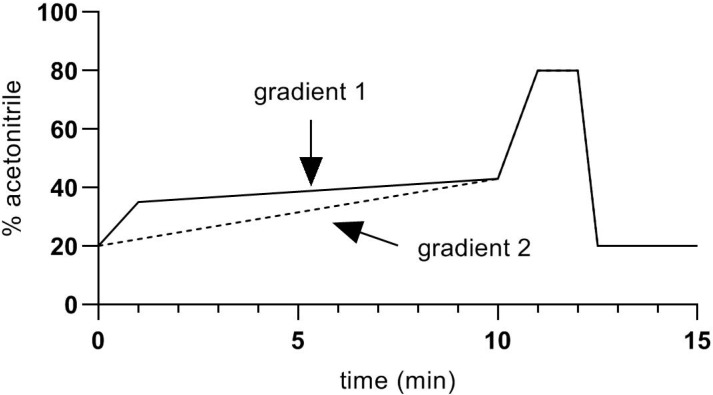
Chromatographic gradients used in the present study.

**Figure 3 animals-10-00821-f003:**
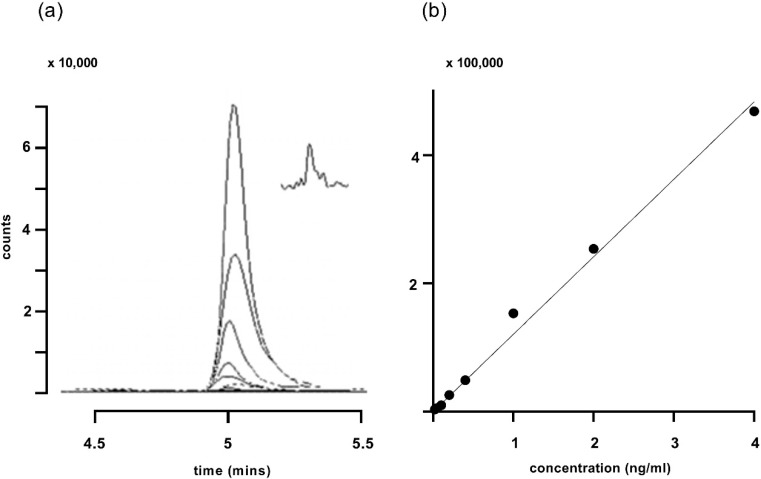
(**a**) Traces for increasing concentrations of corticosterone ranging from 0.02 to 4 ng/mL, using gradient 2. The major peak represents the quantifier ion at m/z 121.1. The inset in (**a**) shows the response to 0.02 ng/mL corticosterone on a compressed time scale. (**b**) A standard line plotting the integrated peak areas (filled circles) from (**a**). R^2^ for the fitted line was 0.994.

**Figure 4 animals-10-00821-f004:**
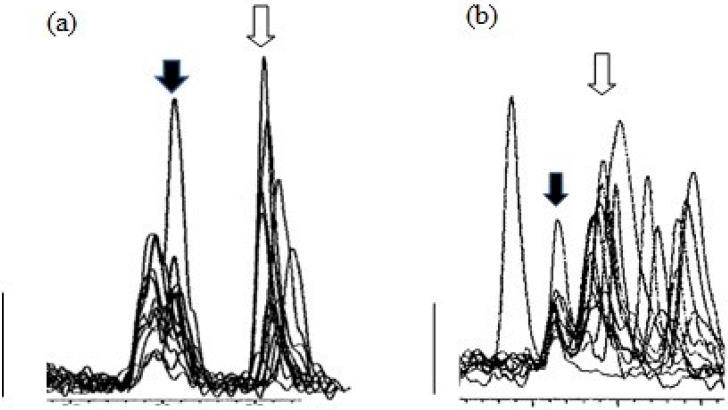
Peaks from MS traces using Gradient 1 (**a**) and Gradient 2 (**b**) for albumen extracts (12) from cage eggs showing the corticosterone peak (filled arrows) and other peaks at similar retention times, including the peaks for 11-deoxy cortisol (open arrows). Major tick marks represent 0.5 min. Bars represent 1000 counts.

**Figure 5 animals-10-00821-f005:**
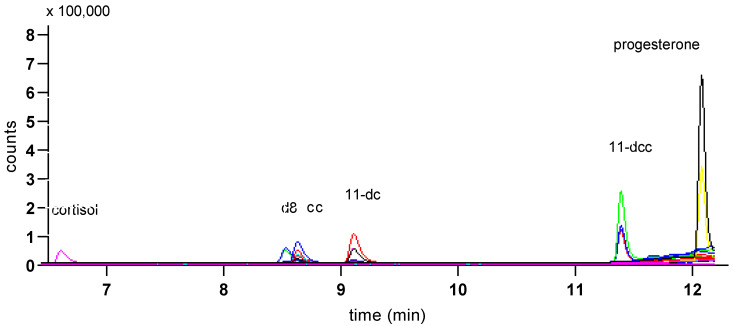
MS trace from a mix of standards of the studied steroids (1 ng/mL) showing peaks for cortisol, d8-corticosterone (d8), corticosterone (cc), 11-deoxycortisol (11-dc), 11-deoxycorticosterone (11-dcc) and progesterone, using Gradient 1. Traces from both quantifier (the larger trace for each peak) and qualifier ions are shown.

**Figure 6 animals-10-00821-f006:**
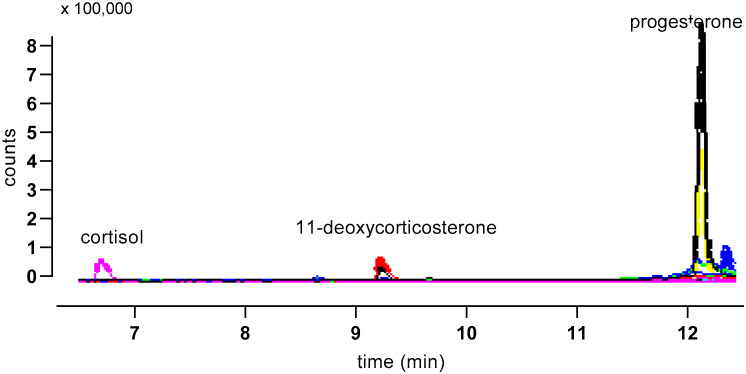
Illustrative MS trace for an albumen extract separated using Gradient 1. Major peaks are indicated, and traces from both quantifier (larger trace for each peak) and qualifier ions are shown.

**Figure 7 animals-10-00821-f007:**
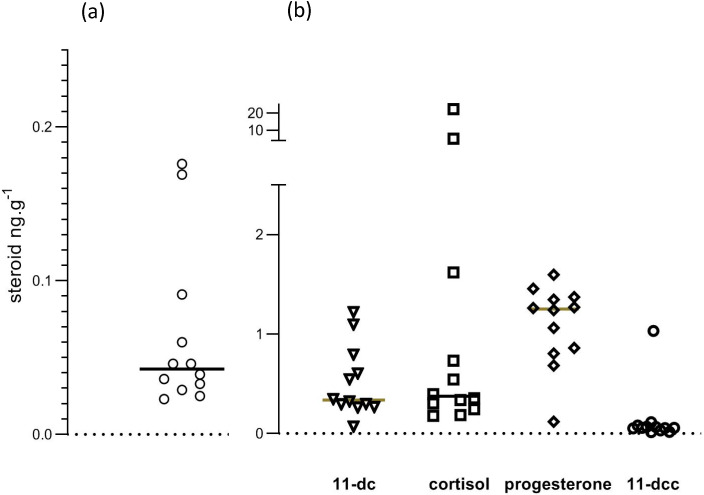
Steroid levels in egg albumen. Values are shown in (**a**) for corticosterone and in (**b**) for 11-deoxy cortisol (11.dc), cortisol, progesterone and 11-deoxy corticosterone (11-dcc).

**Table 1 animals-10-00821-t001:** Multiple reaction monitoring (MRM) transitions and retention times for the studied steroids.

Steroid	Precursor Ion	Quantifier Ion	Qualifier Ion	Retention Time (min)
corticosterone	347.2	121.1	329.3	8.60 *5.01* ^1^
d8 corticosterone	355.2	125.1	337.3	8.49
cortisol	363.2	121.1	91.1	6.54
11-deoxy cc	331.1	97.1	331.1	11.40
11-deoxycortisol	347.1	97.1	109.1	9.08
progesterone	315.1	97.1	109.1	12.10

^1^ Retention times are for gradient 1, except for the italicized time for corticosterone, which was obtained with gradient 2. 11-deoxy cc is 11-deoxycorticosterone.

**Table 2 animals-10-00821-t002:** Median steroid concentrations in egg albumen.

Steroid ^1^	Concentration (ng/g)
corticosterone	0.043 (0.029–0.091)
11-deoxycortisol	0.335 (0.27–0.79)
cortisol	0.374 (0.24–1.60)
11-deoxycorticosterone	0.057 (0.033–0.08)
progesterone	1.251 (0.81–0.84)

^1^ Values are from 12 eggs. Values in brackets are 95% confidence limits of the median; corticosterone values are corrected for recovery.

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
