# Peer review of "HPLC MS-MS Analysis Shows Measurement of Corticosterone in Egg Albumen Is Not a Valid Indicator of Chicken Welfare"

_animals, 2020, doi:10.3390/ani10050821_

Round 1

Reviewer 1 Report

The paper is attempting to do two rather different things, one more successfully than the other. First it aims to show that an HPLC analysis of egg albumen provides a more informative measure of egg corticosterone concentrations (and of many related molecules) than traditional immunoassay measures. In this it appears to be reasonably successful. Second it aims to compare concentrations of these substances from eggs obtained from cage and free-range systems, but the study design for this part is not acceptable and so the conclusions from this part are not justified. 

In revising the manuscript, the authors could consider whether the term "battery cage" is useful. It has been criticised and it is more common to refer to conventional cages in the scientific literature. Not clear why battery cage is used without inverted commas, but 'free range' (line 53) has inverted commas. Better to refer the reader to a source that describes each of these systems clearly. 

Line 53 - there have not been very many studies that have directly compared conventional cage and free range systems - perhaps list them? Sherwin et al., 2010 British Poultry Science for example. 

Line 63 - the 1991 reference is rather old, any updates? 

Line 65 - not justified to say that availability of techniques has driven this research, might be better to say it has facilitated it. 

Line 66 - is there any useful distinction to be made between studies that have used ELISA and those that have used RIA? Do both suffer same problem - text would be clearer if authors could distinguish on this point. 

Line 70 - actually, use of egg corticosterone is not that common and may be restricted to the references already cited. How does this compare to the use of other non-invasive measures e.g. fecal corticosterone? Have there been more studies using this measure? Do any of the same arguments about specificity of assays apply to fecal measures too? 

Line 73 - this is not quite fair to authors of reference 8, who write that measuring stress is informative or relevant for welfare. They don't write directly that equivalent levels of cort in egg albumen indicates equivalent welfare. In fact, their report is far more nuanced, showing larger variation within systems (e.g. different free range farms or different cage farms) than between systems. Also, in linking higher levels of cort to specific events such as outbreaks of ectoparasites. 

Generally, the results here suggest that previous studies may have measured not corticosterone as assumed, but closely-related substances such as cortisol. However, how much does this matter from a welfare perspective? Cortisol is also an indicator of bird stress and if previous studies have shown strong correlations between specific environmental events and egg (supposed) corticosterone increases, then perhaps it could be argued that the immunoassay studies are still valid, but that the authors should stop referring to them as corticosterone assays, and refer instead to corticosterone-family assays?  

Similarly, Cook et al., 2009 found a relationship between plasma corticosterone in hens and egg albumen (supposed) corticosterone the following day. These sorts of results suggest that it may not be necessary to totally discard the use of immunoassay. A more nuanced discussion of this point is needed. 

The most substantial problem with the paper is the comparison between the 12 caged and 12 free-range eggs. Apart from small sample size to draw such a system-level conclusion, there is no information about the characteristics of the flocks from where these eggs were obtained (e.g. flock age, flock breed - both factors known to be highly important in assessing stress level) and there is no flock-level replication. It is therefore not a meaningful comparison and it actually detracts rather than adds to the paper. 

In short the paper would be improved by presenting itself as a study that shows that:

HPLC analysis is more informative than immunoassay in analysis of corticosterone-type materials in eggs. 

Discussion of whether results previously interpreted as "corticosterone" may still have any validity if re-interpreted as "corticosterone-related" - many of the related substances may also play a role in stress response. 

Presents a short cost/ benefit analysis to help other scientists decide which technique to use in future (prep time, kit costs, access to facilities) - just a sentence or two would be helpful. 

Omits the rather spurious system comparison, but suggests how this technique could best be applied in this context. 

Some of the very general points e.g. about conflation of stress and welfare, or about possibility of depletion or exhaustion of stress response could be better integrated with criticism/reinterpretation of previous work on hen housing systems in the discussion. The introduction would be best focussed on the methodological aspects. 

Author response to report 1:
Authors' Notes

Author Response File

This is a very important study to draw attention to the problems of stress assessment using hormonal analyses. I agree with the conclusion of this paper which is to some degree generalizable to other hormonal measurements and their interpretation. There have been papers with this intention before but this deals with the specific case of corticosterone measurements in egg albumen.

My main criticism is the statistical analyses and presentation of the results which could be improved (see below).

Paragraph 86ff: This reads like a summary or conclusion. Instead, I expect the outline of the aim of the study.

Line 115ff: You should provide the number of eggs and if you know, the hybrid. At least you could write whether the eggs were white or brown.

Figure 5: I do not know what the colors mean.

I do not understand the difference between Figures 5 and 6.

Lines 235 – 237: Please provide the data in an appendix or write 'data not shown'.

Lines 263ff and Table 2: Esp. with few data points these tests almost always indicate non-normality. Therefore, it would be better to use a visual evaluation whether the distribution is normal or not. Furthermore, the data points do not need to be normally distributed but the residuals have to be. Did you check the residuals? Using a non-parametric test means that you have less power. Normally, this is the more conservative way. In your case, as you want to show that no difference can be detected, it is the other way round. Besides the P value you should present the test statistic and the confidence intervals.

Response to Reviewer 1; Round 2

Data relating to free range eggs has been deleted, and reference to comparisons between data in cage and free range systems has been removed, as follows:

Lines 20, 41, 94, 96, 125, 203, 240, line 258 = Figure 7, 272, line 278 = Table 2, 280, 285, 294, 301.

Please note also the substitution of the word ‘arguably’ for ‘demonstrably’ at line 318.

Reviewer 2 Report

This is a very important study to draw attention to the problems of stress assessment using hormonal analyses. I agree with the conclusion of this paper which is to some degree generalizable to other hormonal measurements and their interpretation. There have been papers with this intention before but this deals with the specific case of corticosterone measurements in egg albumen.

My main criticism is the statistical analyses and presentation of the results which could be improved (see below).

Paragraph 86ff: This reads like a summary or conclusion. Instead, I expect the outline of the aim of the study.

Line 115ff: You should provide the number of eggs and if you know, the hybrid. At least you could write whether the eggs were white or brown.

Figure 5: I do not know what the colors mean.

I do not understand the difference between Figures 5 and 6.

Lines 235 – 237: Please provide the data in an appendix or write 'data not shown'.

Lines 263ff and Table 2: Esp. with few data points these tests almost always indicate non-normality. Therefore, it would be better to use a visual evaluation whether the distribution is normal or not. Furthermore, the data points do not need to be normally distributed but the residuals have to be. Did you check the residuals? Using a non-parametric test means that you have less power. Normally, this is the more conservative way. In your case, as you want to show that no difference can be detected, it is the other way round. Besides the P value you should present the test statistic and the confidence intervals.

Author response to report 1:
Authors' Notes

Author Response File

Reviewer 3 Report

This study aims to validate the use of immuno-assays for corticosterone in albumin. As stress hormones are typically a major physiological measure of welfare, this study is important to understanding how to measure this parameter. Overall, this is a well written study that has great implications for the use of stress hormones when identifying the welfare status of chickens.

I have two comments:

P2L86-93: This paragraph should state the goals, aims, and/or hypotheses of the present study, not a summary of results and conclusion. Please revise.

How many eggs were sampled for each treatment? Were any samples pooled?

Response to Reviewer 3

Author's Notes

Thank you for your comments

P2L86-93: This paragraph should state the goals, aims, and/or hypotheses of the present study, not a summary of results and conclusion. Please revise.

Apologies. We were responding to indications in the pro forma manuscript, which we now see are not in the instructions to authors. We have revised lines 92-94 accordingly.

How many eggs were sampled for each treatment? Were any samples pooled?

12 eggs were used. This is now stated at lines 124 and 127. Samples were not pooled.

Round 2

Reviewer 1 Report

With the withdrawal of the spurious housing system comparison, the paper is now potentially publishable.

However, the technical/laboratory aspects of the paper should be reviewed by someone with expertise in this area. 

Reviewer 2 Report

I have no further comments to the revised version. All my concerns have been satisfactorily addressed. I consider this study very important.

Reviewer 3 Report

Authors have addressed all comments. Well written manuscript.